# Characterization of Metallo β-Lactamase Producing *Enterobacterales* Isolates with Susceptibility to the Aztreonam/Avibactam Combination

**DOI:** 10.3390/antibiotics13121221

**Published:** 2024-12-17

**Authors:** Brunella Posteraro, Flavio De Maio, Teresa Spanu, Maria Alejandra Vidal Pereira, Francesca Romana Fasano, Maurizio Sanguinetti

**Affiliations:** 1Unità Operativa “Medicina di Precisione in Microbiologia Clinica”, Direzione Scientifica, Fondazione Policlinico Universitario A. Gemelli IRCCS, 00168 Rome, Italy; brunella.posteraro@unicatt.it; 2Dipartimento di Scienze Biotecnologiche di Base, Cliniche Intensivologiche e Perioperatorie, Università Cattolica del Sacro Cuore, 00168 Rome, Italy; 3Dipartimento di Scienze di Laboratorio ed Ematologiche, Fondazione Policlinico Universitario A. Gemelli IRCCS, 00168 Rome, Italy; flavio.demaio@policlinicogemelli.it (F.D.M.); tspanu@gmail.com (T.S.); 4Medical Affairs, Pfizer Italia, 00188 Rome, Italy; mariaalejandra.vidalpereira@pfizer.com (M.A.V.P.); francescaromana.fasano@pfizer.com (F.R.F.)

**Keywords:** metallo-β-lactamase, *Enterobacterales*, antimicrobial resistance, whole genome sequencing, β-lactam/β-lactamase inhibitor combination

## Abstract

**Background/Objectives:** Metallo-β-lactamases (MBLs) in *Enterobacterales* and other Gram-negative organisms pose significant public health threats due to their association with multidrug resistance (MDR). Although aztreonam (AZT) can target MBL-producing organisms, its efficacy is compromised in organisms expressing additional β-lactamases that inactivate it. Combining AZT with the β-lactamase inhibitor avibactam (AVI) may restore its activity against MBL-producing isolates. **Methods:** AZT-AVI, along with other clinically relevant antimicrobials, was tested against thirteen MBL-producing clinical isolates of *Enterobacterales* (nine *Klebsiella pneumoniae*, three *Enterobacter cloacae*, and one *Providencia stuartii*) using whole-genome sequencing (WGS) for genetic characterization. **Results:** AZT-AVI demonstrated full susceptibility across all isolates, whereas aztreonam alone was ineffective. The newer β-lactam/β-lactamase inhibitor combinations imipenem/relebactam and meropenem/vaborbactam were inactive in 100% and 92.3% of isolates, respectively. WGS-based analysis revealed multiple resistance mechanisms consistent with MDR phenotypes, including high-risk *K. pneumoniae* clones (ST147 and ST11). **Conclusions:** AZT-AVI is effective against MDR MBL-producing *Enterobacterales*, highlighting its therapeutic potential for challenging infections. While WGS does not replace phenotypic testing, it provides valuable insights for antimicrobial stewardship and the monitoring of resistance gene dissemination.

## 1. Introduction

Metallo-β-lactamases (MBLs), such as New Delhi MBL (NDM) and Verona integron-encoded MBL (VIM), pose a significant public health threat due to their global spread among *Enterobacterales* and other Gram-negative organisms, with initial isolations in India and Italy, respectively [1]. As acquired enzymes (mostly transmitted by plasmids), MBLs are associated with multidrug resistance (MDR) phenotypes due to their broad-spectrum hydrolysis of β-lactams (excluding the monobactam aztreonam); resistance to multiple aminoglycosides, fluoroquinolones, and other antibiotics; and widespread presence in both environmental and hospital settings [1]. These characteristics make MBLs particularly concerning when compared to serine β-lactamases, including the *Klebsiella pneumoniae* carbapenemase (KPC)-type or oxacillinase-48 (OXA-48)-type β-lactamases [2]. Initially considered as a sparing β-lactam against MBL-producing Gram-negative organisms, aztreonam becomes ineffective with organisms that, although capable of hydrolyzing the drug, express additional β-lactamases that inactivate it [3].

Although β-lactams are highly effective and widely prescribed, their use has been increasingly limited by resistance driven by β-lactamase enzymes. While β-lactamase inhibitors can restore effectiveness against serine β-lactamases, they remain ineffective against the spreading and evolving metallo-β-lactamases (MBLs) [4], which confer resistance to nearly all β-lactams, including carbapenems [3]. New MBL variants continue to emerge rapidly, underscoring the remarkable adaptability of these enzymes, with over 60, 80, and 100 variants identified in the NDM, VIM, and IMP families, respectively [5]. In this context, taniborbactam, a third-generation β-lactamase inhibitor, shows potential against key MBLs, though reduced efficacy against certain NDM and VIM variants, as well as inefficacy against IMP-type MBLs, tempers enthusiasm [6].

Currently, the second-generation β-lactamase inhibitor avibactam (AVI), when combined with aztreonam (AZT) to inhibit non-MBL β-lactamases [6], enables aztreonam to regain efficacy against MBL-producing Gram-negative organisms, particularly NDM-producing *Enterobacterales* [7]. These findings are consistent with data from the 2016–2020 ATLAS Global Surveillance Program study (106,686 *Enterobacterales* clinical isolates from 63 countries), showing that 98.1% of isolates with different NDM variants and 99.7% of MBL-producing isolates were inhibited by AZT-AVI at ≤8 mg/L (the provisional pharmacokinetic/pharmacodynamic breakpoint available in 2023, the study’s publication year) [8]. In 2022, Le Terrier et al. [9] studied a collection of 44 MDR MBL-producing *Enterobacterales* clinical isolates and reported that the susceptibility rate for AZT-AVI (70.3%) was lower than that for aztreonam/taniborbactam (75.0%), both of which were significantly lower than the rate observed for aztreonam combined with zidebactam (98.4%), a recently developed β-lactamase inhibitor. In April 2024, the European Medicines Agency (EMA) approved AZT-AVI for patients with complicated intra-abdominal infection; hospital-acquired pneumonia, including ventilator-associated pneumonia; complicated urinary tract infection, including pyelonephritis; and infections caused by aerobic Gram-negative organisms with limited treatment options (https://ec.europa.eu/health/documents/community-register/html/h1808.htm, accessed on 10 December 2024). In May 2024, the European Committee on Antimicrobial Susceptibility Testing (EUCAST) set the clinical breakpoint at a minimum inhibitory concentration (MIC) of 4 mg/L (https://www.eucast.org/fileadmin/src/media/PDFs/EUCAST_files/Rationale_documents/Aztreonam-avibactam_Rationale_Document_v_1.0_20240703.pdf, accessed on 10 December 2024).

Here, we report on the full susceptibility of AZT-AVI against genetically well-characterized *Enterobacterales* clinical isolates producing NDM and VIM carbapenemases. The whole genome sequencing analysis revealed the presence of various antimicrobial resistance genes, consistent with a MDR phenotype shown in vitro for all isolates. As expected, none of the isolates proved to be susceptible to the β-lactam/β-lactamase inhibitor combinations currently included in antimicrobial susceptibility testing assays.

## 2. Results and Discussion

We studied 13 MBL-producing *Enterobacterales* isolates from an observational, prospective, multicenter study on the epidemiology, management, and outcomes of carbapenem-resistant *Enterobacterales* (and *Pseudomonas aeruginosa*) infections in intensive care unit patients across Italian hospitals (INCREASE-IT study). Of these thirteen isolates, nine were identified as *K. pneumoniae* (Kp1 to Kp9), three as *Enterobacter cloacae* (Ec1 to Ec3), and one as *Providencia stuartii* (Ps1). Regarding MBLs, eight (61.5%) of the thirteen isolates produced an NDM-type carbapenemase, specifically NDM-1 (six in *K. pneumoniae*, one in *E. cloacae*, and one in *P. stuartii*). The remaining five (38.5%) isolates (three *K. pneumoniae* and two *E. cloacae*) produced a VIM-type carbapenemase, specifically VIM-1. In two (40.0%) of the five VIM-1-producing isolates (*K. pneumoniae* Kp6 and Kp9), VIM-1 was found in association with a KPC-type carbapenemase, namely KPC-2 and KPC-3, respectively.

As shown in Table 1, only the *P. stuartii* isolate (Ps1) was classified as susceptible, increased exposure (MIC, 4 mg/L), while all nine *K. pneumoniae* and three *E. cloacae* isolates were classified as resistant to aztreonam (MICs, ≥32 mg/L), according to the 2024 EUCAST clinical breakpoints [10]. In contrast, all 13 isolates (100%) were classified as susceptible to AZT-AVI (MICs, 0.03 to 0.5 mg/L) using the EUCAST breakpoints (susceptible: MIC ≤ 4 mg/L; resistant: MIC > 4 mg/L) mentioned above.

Regarding the antimicrobials routinely tested in a clinical microbiology laboratory, all 13 isolates (100%) were resistant to carbapenems (ertapenem, meropenem, and/or imipenem), third-generation cephalosporins (cefepime, cefotaxime, ceftazidime, ceftazidime/avibactam, and ceftolozane/tazobactam), penicillins (amoxicillin/clavulanic acid and piperacillin/tazobactam), and fluoroquinolones (ciprofloxacin). These findings align with an MDR phenotype, defined as non-susceptibility to at least one antimicrobial agent in three or more antimicrobial classes [11]. Additionally, 11 of the 13 isolates (84.6%) were resistant to trimethoprim/sulfamethoxazole, and 5 (38.5%) were resistant to aminoglycosides (amikacin and/or gentamicin).

Among the newer β-lactam/β-lactamase inhibitor combinations tested in the clinical microbiology laboratory, imipenem/relebactam and meropenem/vaborbactam were inactive in 13 (100%) and 12 (92.3%) of the 13 isolates, respectively. The isolate (Kp4; VIM-1 positive) with an MIC of 8 mg/L for meropenem/vaborbactam, which falls within the EUCAST susceptible (S) breakpoint (≤8 mg/L) [10], also had an MIC of 8 mg/L for meropenem alone, classifying it in the EUCAST susceptible, increased exposure category for meropenem [10]. Regarding cefiderocol and colistin, which are considered the two last-resort antimicrobial agents available to date [3], ten (76.9%) and eleven (84.6) of the thirteen isolates were susceptible, respectively. Of the two colistin-resistant isolates (Ps1 and Kp5, both NDM-1 positive), the Kp5 isolate was also found to be resistant to cefiderocol.

Initially, we analyzed whole-genome sequencing (WGS) data from the study isolates to confirm their identity, verify the presence of MBL-encoding genes in all isolates (as detailed above), and determine the sequence type (ST) for the nine *K. pneumoniae* isolates. The most frequent STs were ST147, found in four (44.5%) isolates (Kp1, Kp3, Kp7, and Kp8), and ST11, found in two (22.2%) isolates (Kp2 and Kp5). The remaining three *K. pneumoniae* isolates (11.1% each) belonged to ST1876 (Kp4), ST17 (Kp6), and ST307 (Kp9). Both ST147 and ST11 are high-risk *K. pneumoniae* clones whose spread in hospital settings should be actively controlled [12].

We then explored the distribution of antimicrobial resistance genes/determinants among the 13 isolates included in the study (Figure 1). The resistome profiles obtained for the nine *K. pneumoniae* (Kp1 to Kp9), three *E. cloacae* (Ec1 to Ec3), and one *P. stuartii* (Ps1) isolates primarily focused on the antimicrobial agents for which MIC results were available. These profiles revealed the presence of either genes encoding broad-spectrum β-lactamases, such as CTX-M-15 (detected in all isolates except Kp6, Ec1, Ec3, and Ps1), OXA-type (in the Kp6 isolate), SHV-12 (in Ec1 and Ec3), and CMY-6 (in Ps1). Additionally, the profiles provided insight into non-β-lactam antimicrobial resistance genes or mutations, further indicating an MDR phenotype.

We found that all isolates, except Ps1, had at least one gene conferring resistance to quinolones, while the *aac(6′)-Ib-cr5* gene, encoding an aminoglycoside acetyltransferase active against fluoroquinolones (ciprofloxacin in this study), was detected only in the Kp5 isolate. Among the 21 aminoglycoside-resistance genes detected, we observed that *aac(2′)-Ia*, *aac(6′)-Ib3*, and *rmtC* were uniquely present in the Ps1 isolate (resistant to both amikacin and gentamicin), while *aac(3)-IIe* and *aac(6′)*-*Ib* were shared by only two isolates, one of which (Kp2) was resistant and the other (Ec2) was susceptible to both amikacin and gentamicin. Besides Ec2, there were four other aminoglycoside-susceptible isolates (Kp4, Kp6, Kp8, and Ec3), with Ec3 carrying five genes (*aac(6′)-Ib4*, *aadA1*, *aph(3″)-Ib*, *aph(3′)-XV*, and *aph(6)-Id*), Kp4 carrying four genes (*aac(6′)-Ib4*, *aadA1*, *aph(3″)-Ib*, and *aph(6)-Id*), and Kp6 and Kp8 each carrying three genes (*aac(6′)-Ib4*, *aadA1*, and *aph(3′)-XV*, and *aadA1*, *aph(3′)-Ia*, and *aph(3′)-VI*, respectively) genes detected. Regardless of aminoglycoside-susceptible or -resistant phenotypes, *aadA1* was detected in most isolates (8/13, 61.5%).

Conversely, in two isolates (Kp1 and Ps1) found to be susceptible to trimethoprim/sulfamethoxazole, we did not detect any of the three resistance genes found in the remaining eleven resistant isolates (*dfrA1* in one isolate, *dfrA14* in seven isolates, and *dfrA6* in three isolates). Lastly, we detected the colistin-resistance-associated mutation *pmrB_R256G* [13] in six of the thirteen isolates (Kp1 to Kp3, Kp5, and Kp7 to Kp8), including Kp5, one of the two colistin-resistant isolates in the study.

These findings underscore the complexity of antimicrobial resistance mechanisms in *K. pneumoniae* and other *Enterobacterales* organisms, highlighting the need for further research to clarify the relationship between genotypes and phenotypes in these clinically significant organisms. Notably, we could not discuss WGS data regarding macrolides, phenicols, or sulfonamides, as susceptibility testing for these agents was not conducted. Similarly, we did not examine cefiderocol resistance in three of the thirteen isolates (Kp5, Kp7, and Ec3) where it was detected via disk diffusion, as cefiderocol-resistance mechanisms (such as *cirA* gene alterations in NDM-1-producing *K. pneumoniae*) remain only partially understood [14].

Additionally, WGS analysis was performed to investigate the phylogenetic relationships among the nine *K. pneumoniae* isolates included in the study (Figure 2).

As depicted in Figure 2a, Kp1, Kp3, and Kp8 formed a cluster, with Kp1 differing from Kp3 by 13 alleles and from Kp8 by 27 alleles. Kp7 was more closely related to Kp3, differing by 23 alleles. The remaining five isolates were highly distant from this cluster and, apart from Kp2 and Kp5 (which differed by only 22 alleles), were also distant from each other. As depicted in Figure 2b, the matrix of genomic features, including plasmids and mobile genetic elements, is organized according to the core genome phylogeny based on single-nucleotide polymorphisms. Together, these data suggest that the level of evolutionary relatedness among the *K. pneumoniae* isolates in this study reflects the distinct distribution and transfer profiles of clinically relevant bacterial genes.

This study has limitations, including the small number of isolates and their collection from a single geographical region, which may limit the generalizability of the findings.

## 3. Methods

### 3.1. Study Isolates

This study was conducted at the clinical microbiology laboratory of a large tertiary-care teaching hospital in Rome, Italy, using clinical isolates of carbapenem-resistant *Enterobacterales* (9 *Klebsiella pneumoniae*, 3 *Enterobacter cloacae*, and 1 *Providencia stuartii*) selected from a collection of isolates enrolled in the INCREASE-IT study. All isolates that satisfied the inclusion criteria were included in the study. Isolates were selected if they exhibited in vitro carbapenem resistance due to the presence of MBLs, as determined by either multiplex PCR identification (BCID2, bioMérieux, Marcy l’Étoile, France) or lateral flow immunoassay (NG-Test CARBA 5, NG Biotech, Guipry-Messac, France). No additional isolates in the collection met these predefined criteria, ensuring that all eligible isolates were included. Access to the origin of the isolates (e.g., hospital, country) and to the personal data of the patients from whom the isolates originated was not permitted. Before use, isolates provided by each center to the laboratory as frozen glycerol stocks were revitalized and checked for purity by culturing at 37 °C on 5% sheep blood agar (SBA) plates. This step ensured that all isolates were viable and free from contamination prior to further analysis.

### 3.2. Antimicrobial Susceptibility Testing

For each isolate, MICs for amikacin, amoxicillin/clavulanic acid, aztreonam, cefepime, cefotaxime, ceftazidime, ceftazidime/avibactam, ceftolozane/tazobactam, colistin, ertapenem, imipenem, meropenem, piperacillin/tazobactam, gentamicin, ciprofloxacin, and trimethoprim/sulfamethoxazole were determined by the broth microdilution method (BMD) using antimicrobial drug-containing MDRO plates (Bruker Daltonics, Bremen, Germany). In parallel, MICs for imipenem/relebactam and meropenem/vaborbactam were determined by the Etest method (bioMérieux), and MICs for AZT-AVI were determined by the gradient strip diffusion method (an Etest-equivalent method) using MTS (MIC Test Strip; Liofilchem, Roseto Degli Abruzzi, Teramo, Italy), according to the manufacturers’ instructions. The FDA-cleared MTS method was recently shown to be accurate for AZT-AVI MIC determination in MBL-producing *Enterobacterales* [15]. To ensure comparability with BMD results, MICs between two 2-fold dilutions were rounded up to the higher MIC value. Susceptibility to cefiderocol was assessed by the disk diffusion method using 30 μg disks provided by Liofilchem, as described elsewhere [16] and in accordance with EUCAST guidelines [17]. Quality control strains *Escherichia coli* ATCC 25922, *E. coli* ATCC 35218, and *K. pneumoniae* ATCC 700603 were used in each run. MICs were interpreted according to EUCAST breakpoints v14.0 [10]. Unlike EUCAST, the Clinical and Laboratory Standards Institute (CLSI) has not established specific breakpoints for AZT-AVI. However, for validation purposes, we also interpreted the MICs for AZT-AVI of the study isolates using the CLSI aztreonam breakpoints (susceptible: MIC ≤ 4 mg/L; intermediate: MIC 8 mg/L; resistant: MIC ≥ 16 mg/L). Based on these breakpoints, all isolates were classified as susceptible to AZT-AVI.

### 3.3. Whole Genome Sequencing

All WGS experiments were performed according to previously established procedures [18]. DNA from all 13 isolates was extracted using the DANAGENE Microbial DNA kit (Danagen-Bioted, Barcelona, Spain), and concentration and purity were assessed with a NanoDrop One spectrophotometer (Thermo Fisher, Waltham, MA, USA). Paired-end short reads were generated starting from DNA libraries with the Illumina DNA Prep kit (Illumina, San Diego, CA, USA) and subsequent sequencing on an Illumina MiSeq DX platform following Illumina’s recommendations. Raw reads were quality controlled and trimmed using the fastp tool (https://github.com/OpenGene/fastp, accessed on 10 December 2024) and then assembled using the Unicycler pipeline (https://github.com/rrwick/Unicycler, accessed on 10 December 2024). The identity of all 13 isolates was confirmed by matching against the KmerFinder database (https://bitbucket.org/genomicepidemiology/kmerfinder_db/src/master/, accessed on 10 December 2024).

For 9 of the 13 study isolates (all *K. pneumoniae*, Kp1 to Kp9), short-read assemblies were used to determine the ST using a seven-loci multilocus sequence typing (MLST) scheme [19] and to create a core-genome MLST scheme, based on 2358 genes, as implemented in Ridom SeqSphere+, as described elsewhere [20]. For these 9 isolates, a minimum spanning tree, based on the core-genome MLST profiles, and a neighbor-joining tree, based on core-genome single nucleotide polymorphisms, were generated.

For each of the 13 isolates, WGS data were analyzed with both AMRFinderPlus (https://github.com/ncbi/amr, accessed on 10 December 2024) and ABRicate (https://github.com/tseemann/abricate), which enabled the identification of antimicrobial resistance genes/determinants. A matrix was created to depict the distribution of antimicrobial resistance genes and point mutations across the 13 isolates included in the study. Plasmids and mobile genetic elements were identified using PlasmidFinder (https://github.com/genomicepidemiology/plasmidfinder, accessed on 10 December 2024) and MGEfinder (https://github.com/bhattlab/MGEfinder, accessed on 10 December 2024), respectively, and their distribution was represented in a separate matrix focusing on the 9 *Klebsiella pneumoniae* isolates. All raw sequence data have been deposited in the NCBI Sequence Read Archive (BioProject accession number: PRJNA1187234).

## 4. Conclusions

In summary, our findings confirm the full activity of the AZT-AVI combination against MDR MBL-producing *Enterobacterales*, underscoring its potential as a therapeutic option for challenging infections. This study also highlights the value of WGS as a robust tool for the genetic characterization of MDR pathogens [21]. Although WGS does not replace phenotypic testing, it provides valuable insights to support rational antimicrobial use and monitor resistance gene spread, particularly in hospital and ICU settings. Further research is needed to deepen our understanding of genotype–phenotype correlations, especially for emerging resistance mechanisms like those linked to cefiderocol. Integrating WGS with conventional methods could therefore enhance therapeutic and infection control strategies in clinical practice.

## Figures and Tables

**Figure 1 antibiotics-13-01221-f001:**
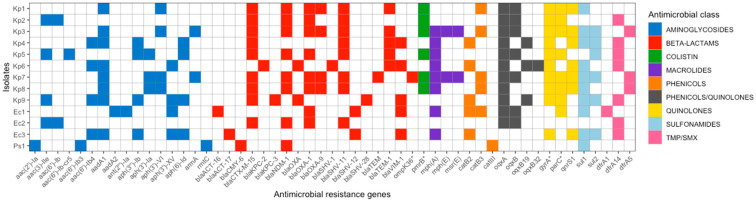
Mapping of antimicrobial resistance genes/determinants across the bacterial isolates (*n* = 13) included in the study. All isolates exhibited multiple antimicrobial resistance mechanisms. Colored blocks indicate the presence of a gene/determinant associated with resistance to specific classes of antimicrobial agents. Asterisks denote mutated genes. Detected point mutations include those associated with resistance to carbapenems (*ompK36_G133D*), colistin (*pmrB_R256G*), and fluoroquinolones (*gyrA_S83F*, *gyrA_S83I, gyrA_S83Y*, *gyrA_D87A*, *gyrA_D87N*, and *parC_S80I*). TMP/SMX, trimethoprim/sulfamethoxazole.

**Figure 2 antibiotics-13-01221-f002:**
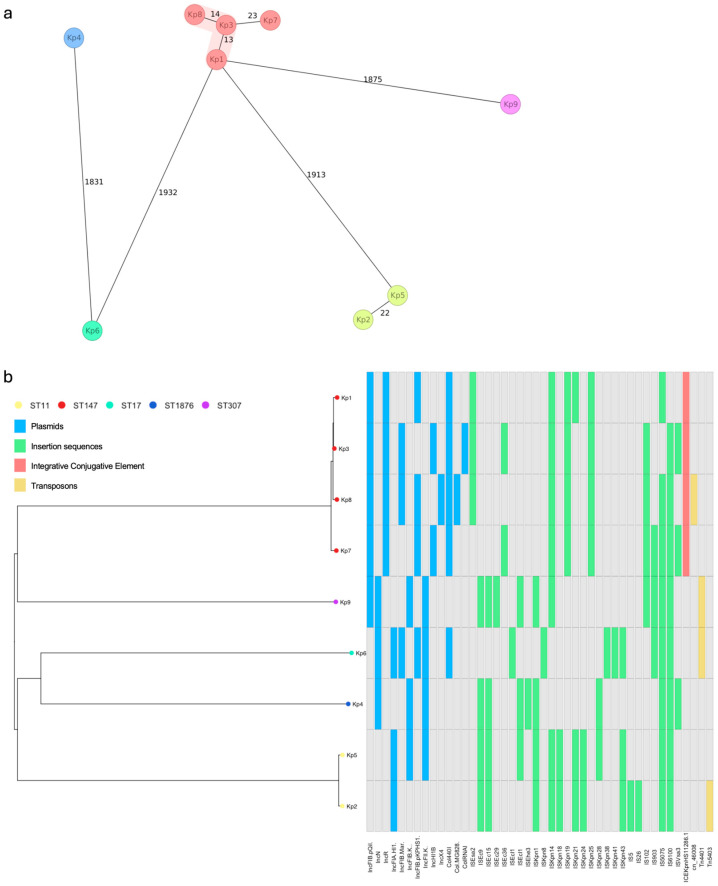
Core genome phylogeny of nine *K. pneumoniae* isolates (Kp1 to Kp9) sequenced in this study. (**a**) The MLST-based minimum spanning tree shows the relatedness among the isolates, represented by colored circles at the tree nodes. Numbers indicate allele differences between nodes. (**b**) The SNP-based neighbor-joining tree shows the relatedness among the isolates, represented by colored circles (colors correspond to the different STs identified) at the terminal nodes of the tree. The tree is organized according to the presence (colored) or absence (gray) of matrix-distributed genomic features, such as plasmids and mobile genetic elements. These include insertion sequences, integrative conjugative elements, and transposons. Kp, *Klebsiella pneumoniae*; MLST, multilocus sequence typing; SNP, single nucleotide polymorphism; ST, sequence type.

**Table 1 antibiotics-13-01221-t001:** Susceptibility testing results of aztreonam and aztreonam/avibactam combination for 13 *Enterobacterales* clinical isolates.

Species (Isolate) Tested	Results of Aztreonam Expressed as:		Results of Aztreonam/Avibactam Expressed as:	
	MIC (mg/L)	Interpretive Category	MIC (mg/L)	Interpretive Category
*K. pneumoniae* (Kp1)	≥32	Resistant	0.25	Susceptible
*K. pneumoniae* (Kp2)	≥32	Resistant	0.25	Susceptible
*K. pneumoniae* (Kp3)	≥32	Resistant	0.12	Susceptible
*K. pneumoniae* (Kp4)	≥32	Resistant	0.03	Susceptible
*K. pneumoniae* (Kp5)	≥32	Resistant	0.25	Susceptible
*K. pneumoniae* (Kp6)	≥32	Resistant	0.25	Susceptible
*K. pneumoniae* (Kp7)	≥32	Resistant	0.25	Susceptible
*K. pneumoniae* (Kp8)	≥32	Resistant	0.12	Susceptible
*K. pneumoniae* (Kp9)	≥32	Resistant	0.25	Susceptible
*E. cloacae* (Ec1)	≥32	Resistant	0.5	Susceptible
*E. cloacae* (Ec2)	≥32	Resistant	0.12	Susceptible
*E. cloacae* (Ec3)	≥32	Resistant	0.25	Susceptible
*P. stuartii* (Ps1)	4	Susceptible, increased exposure	0.5	Susceptible

MIC, minimum inhibitory concentration; Kp, *Klebsiella pneumoniae*; Ec, *Enterobacter cloacae*; Ps; *Providencia stuartii*. Note: For each isolate tested, MICs for aztreonam (AZT) alone or in combination with avibactam (AVI) were determined using broth microdilution- and gradient strip diffusion-based methods, respectively. Based on MIC values for AZT or AZT-AVI, isolates were classified as resistant, susceptible, increased exposure, or susceptible as described in the text. The gradient strip diffusion method, often referred to by the trade name Etest, was used in this study to assess susceptibilities to other antimicrobials.

## Data Availability

Data may be available upon reasonable request.

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
