# Peer review of "Characterization of Metallo β-Lactamase Producing Enterobacterales Isolates with Susceptibility to the Aztreonam/Avibactam Combination"

_antibiotics, 2024, doi:10.3390/antibiotics13121221_

Round 1

Reviewer 1 Report

Comments and Suggestions for Authors

Comments are provided in the attached manuscript pdf. 

Author Response

We thank the reviewer for appreciating our study and providing valuable suggestions to improve the manuscript. Below, we address each comment in detail

Comment 1: 1. No R, I and S were mentioned in the table text. Omit these letters.

Reply: As suggested, we have omitted these letters from the table text. See page 3 of the revised manuscript.

Comment 2: Citations required.

Reply: As required, we have added relevant references to address the dissemination of high-risk Klebsiella pneumoniae clones. See page 4 of the revised manuscript.

Comment 3: Is it short-read Pair End or Single-End? Please mention.

Reply: We have added this specification to the text as requested. See page 7 of the revised manuscript.

Comment 4: Which house-keeping genes were used for MLST typing? Please cite the relevant article.

Reply: Additional information and relevant references have been included to specify the MLST typing scheme. See page 7 of the revised manuscript.

Comment 5: Is this result part necessary in the methodology section? Suggested to shift this information to the appropriate section.

Reply: Following the reviewer’s suggestion, we have shifted the information to the appropriate section. See page 4 of the revised manuscript.

Comment 6: NCBI accession ID is not accessible; kindly release it to the public.

Reply: The NCBI accession ID has now been made publicly accessible.

Reviewer 2 Report

Comments and Suggestions for Authors

The manuscript by Posteraro et al. presents a study on the therapeutic potential of the aztreonam/avibactam (AZT-AVI) combination to treat 13 metallo β-lactamase (MBL) producing Enterobacterales isolates collected from intensive care unit patients. The authors further performed whole-genome sequencing (WGS) to investigate the resistome profiles of these Enterobacterales isolates, confirming the multidrug resistance (MDR) phenotypes of these 13 strains. Although this study has limitations, including 1) the small number of testing strains (13) that may limit the generalizability of the findings, and 2) the strains were isolated from the Italian hospitals, which may not represent the global distribution of MBL-producing Enterobacterales, the manuscript could be accepted as communication because it shows the effectiveness of AZT-AVI against serious MDR pathogens and provides a feasible treatment option against the rising threat of multidrug-resistant (MDR) bacteria.

The manuscript is generally well-written, but some sentences could benefit from language editing to improve clarity, particularly lines 49-53 and lines 68-71. Besides, in line 172, “dfA14” should be “dfrA14”.

Author Response

We thank the reviewer for their positive evaluation of our manuscript and for recognizing the therapeutic potential of the AZT-AVI combination in treating MDR MBL-producing Enterobacterales. Below, we address the reviewer’s specific comments.

Comment 1: Limitations of the study (small number of strains and geographical focus).

Reply: We acknowledge the limitations of the study, including the relatively small number of isolates (13) and their collection from Italian hospitals, which may restrict the generalizability of the findings. These limitations have been explicitly addressed in the revised Discussion section to provide appropriate context for the study’s scope and conclusions. See page 6 of the revised manuscript.

Comment 2: Language editing for clarity (lines 49-53 and lines 68-71).

Reply: We have carefully revised the manuscript, particularly the sentences in lines 49-53 and 68-71, to improve clarity and readability. The updated text ensures that the content is conveyed more effectively.

Comment 3: Typographical error (line 172, “dfA14” should be “dfrA14”).

Reply: Thank you for pointing this out. The typographical error has been corrected, and “dfA14” has been replaced with “dfrA14”.

Reviewer 3 Report

Comments and Suggestions for Authors

I commend the authors for their valuable work. Before this manuscript can be considered for publication, I have the following questions and suggestions for the authors.

Major Comments:

Methods:

  1. Were all isolates equally likely to be selected?
  2. Could the authors clarify how the viability of the isolates was confirmed? Also, did they implement any specific quality control measures to ensure the consistency of the isolates?
  3. The authors have interpreted the MIC using EUCAST guidelines. For validation purposes, could they consider cross-validating some samples using other globally recognized guidelines?

Minor Comments:

  1. In the title, the word "Enterobacterales" is incorrectly hyphenated. Additionally, according to a recent article in the Antibiotics journal, all key words in the title should begin with a capital letter. Therefore, I recommend revising the title as follows: “Characterization of Metallo-β-Lactamase Producing Enterobacterales Isolates with Susceptibility to the Aztreonam/Avibactam Combination.”
  2. Lines 49-52: The sentence is lengthy and may be difficult to read and understand. I suggest breaking it into two or more sentences for better clarity.

Author Response

We sincerely thank the reviewer for their positive feedback and constructive suggestions. Below, we address each comment

Major Comments

Comment 1: Were all isolates equally likely to be selected?

Reply: The selection of isolates was based on predefined criteria, ensuring a focus on MBL-producing Enterobacterales associated with carbapenem resistance. These criteria were applied consistently across all isolates included in the study. To clarify this process, we have added a detailed explanation in the Methods section. See page 6 of the revised manuscript.

Comment 2: Could the authors clarify how the viability of the isolates was confirmed? Also, did they implement any specific quality control measures to ensure the consistency of the isolates?

Reply: Viability was confirmed by subculturing the frozen isolates on 5% sheep blood agar plates at 37°C to check for growth and purity. To ensure consistency, we used standard quality control procedures, including testing well-characterized strains (Escherichia coli ATCC 25922, E. coli ATCC 35218, and Klebsiella pneumoniae ATCC 700603) in each experimental run. These details have now been included in the Methods section for clarity. See pages 6 and 7 of the revised manuscript.

Comment 3: The authors have interpreted the MIC using EUCAST guidelines. For validation purposes, could they consider cross validating some samples using other globally recognized guidelines?

Reply: While EUCAST breakpoints were used as the primary interpretive criteria due to their widespread adoption in Europe and relevance to our study context, we acknowledge the value of cross-validation with other guidelines, such as those from CLSI. To address this, we have added a sentence specifying: “For validation purposes, we also interpreted the MICs for AZT-AVI of the study isolates using the CLSI aztreonam breakpoints (susceptible: MIC ≤4 mg/L; intermediate: MIC 8 mg/L; resistant: MIC ≥16 mg/L). Based on these breakpoints, all isolates were classified as susceptible to AZT-AVI.” See page 7 of the revised manuscript.

Minor comments

Comment 1: In the title, the word “Enterobacterales” is incorrectly hyphenated. Additionally, according to a recent article in the Antibiotics journal, all key words in the title should begin with a capital letter. Therefore, I recommend revising the title as follows: “Characterization of Metallo-β-Lactamase Producing Enterobacterales Isolates with Susceptibility to the Aztreonam/Avibactam Combination.”

Reply: Thank you for pointing this out. We have corrected the hyphenation and revised the title to ensure consistency with the recommended style. The updated title is: “Characterization of Metallo-β-Lactamase Producing Enterobacterales Isolates with Susceptibility to the Aztreonam/Avibactam Combination.”

Comment 2:  Lines 49-52: The sentence is lengthy and may be difficult to read and understand. I suggest breaking it into two or more sentences for better clarity.

Reply: We appreciate this suggestion and have revised the sentence in lines 49-52 by splitting it into two shorter, clearer sentences. This change improves readability and ensures the information is conveyed more effectively. See page 2 of the revised manuscript

Reviewer 4 Report

Comments and Suggestions for Authors

The authors aimed to characterize metallo-β-lactamase-producing Enterobacterales isolates with susceptibility to the aztreonam/avibactam combination. While the study provides valuable and promising information from both scientific and practical perspectives, the manuscript requires minor revisions to enhance its clarity and completeness.

Comments:

  1. The authors have not detailed how the 13 sequences were assembled or specified the bioinformatics platform used for the assembly. Please provide this information in the methods section.

  2. Clarify whether any bioinformatics tools were employed to determine the taxonomy of the 13 isolates. It would strengthen the manuscript to include details about coverage and matching against a specific taxonomy database in the results section.

  3. If possible, generating a core genome SNP tree for the nine Klebsiella isolates would provide insight into their evolutionary relatedness. This would significantly enhance the phylogenetic analysis of the study.

  4. Did the authors identify different plasmid incompatibility groups by analyzing the sequences using plasmid-finding tools, Including information about common plasmids, associated resistance determinants, and attached mobile genetic elements would enrich the discussion.

  5. The manuscript requires careful proofreading for language and grammatical corrections to improve readability and ensure scientific clarity.

Author Response

We thank the reviewer for their valuable feedback and constructive suggestions, which will enhance the clarity and completeness of our manuscript. Below, we address each point in detail.

Comment 1: The authors have not detailed how the 13 sequences were assembled or specified the bioinformatics platform used for the assembly. Please provide this information in the methods section.

Reply: Thank you for this observation. We have updated the Methods section to include details on the bioinformatics tools used for the assembly of sequences. See page 7 of the revised manuscript.

Comment 2: Clarify whether any bioinformatics tools were employed to determine the taxonomy of the 13 isolates.

Reply: We appreciate the reviewer’s suggestion. The taxonomy of the 13 isolates was confirmed by matching their sequences against the KmerFinder database. This information has been added to the Methods section, and a brief sentence addressing this point has been included in the Results section. See pages 4 and 7 of the revised manuscript.

Comment 3: If possible, generating a core genome SNP tree for the nine Klebsiella isolates would provide insight into their evolutionary relatedness.

Reply: We appreciate this excellent suggestion. In response, we have generated a core genome SNP tree for the nine Klebsiella pneumoniae isolates. This phylogenetic tree has been included in a new figure (Figure 2), which also features a core genome MLST tree for the same isolates. These additions have been referenced in the Results and Discussion sections to emphasize the evolutionary relationships among the isolates. See pages 5, 6, and 7 of the revised manuscript.

Comment 4: Did the authors identify different plasmid incompatibility groups by analyzing the sequences using plasmid-finding tools?

Reply: Thank you for raising this important point. We have now analyzed the sequences using PlasmidFinder and MGEfinder to identify plasmids and mobile genetic elements. Details regarding these additional genomic features have been incorporated into the Results and Discussion sections, enhancing the genomic characterization of the isolates. See pages 5, 6, and 7 of the revised manuscript.

Comment 5: The manuscript requires careful proofreading for language and grammatical corrections to improve readability and ensure scientific clarity.

Reply: We have carefully proofread the manuscript to address grammatical issues and enhance clarity. Additionally, several sentences have been revised to improve readability and ensure a smoother flow. We are confident that these revisions have significantly enhanced the overall quality of the manuscript.

Reviewer 5 Report

Comments and Suggestions for Authors

1-  The topic of the manuscript is important, but not novel. After searching the internet, I found well over 30 papers with the same topic, and conclusions.

2-  The conclusions presented are the result of one  Figure (Figure 1). Although the figure is well done, it is not sufficient to make conclusions. 

3-  Table 1, "susceptibility", is routine

4-  This is an important topic and should be worth publishing, but it also needs some additional information. 

5-  I would like to see new and additional information, that would bring it to a new level.

Author Response

We would like to thank the reviewer for their thoughtful comments and for acknowledging the importance of the topic. Below, we address each point in detail.

Comment 1: The topic of the manuscript is important, but not novel. After searching the internet, I found well over 30 papers with the same topic, and conclusions.

Reply: We appreciate the reviewer’s observation. While the general topic of AZT-AVI activity against MBL-producing Enterobacterales has been previously explored, our study provides new and distinct contributions. Specifically, we performed phylogenetic analyses of the Klebsiella pneumoniae isolates, highlighting the predominance of high-risk clones such as ST147 and ST11. In addition, we included genomic details, such as plasmid content, insertion sequences, and antimicrobial resistance gene profiles, which enrich the understanding of resistance mechanisms in MBL-producing isolates. These findings extend beyond routine susceptibility testing and address specific requests made by other reviewers, thereby enhancing the novelty and value of our manuscript.

Comment 2: The conclusions presented are the result of one Figure (Figure 1). Although the figure is well done, it is not sufficient to make conclusions.

Reply: We agree that conclusions should be supported by robust data. While Figure 1 visually summarizes the resistome profiles of the isolates, it complements rather than solely underpins our conclusions. In response to this and other reviewers’ comments, we have added a new figure (Figure 2) that extends the genetic characterization of our isolates through phylogenetic analysis. Together, these figures, combined with comprehensive antimicrobial susceptibility testing and detailed genomic analysis using whole-genome sequencing (WGS), provide a robust basis for our conclusions and enrich the overall findings of the study.

Comment 3: Table 1, “susceptibility”, is routine.

Reply: We acknowledge that susceptibility testing is a standard approach; however, its inclusion in our study serves as a foundation for comparing AZT-AVI activity with other β-lactam/β-lactamase inhibitor combinations and last-resort agents like colistin and cefiderocol. The detailed MIC data and their interpretation using EUCAST breakpoints add clinical relevance, particularly in the context of MDR infections.

Comment 4: This is an important topic and should be worth publishing, but it also needs some additional information.

Reply: We acknowledge that susceptibility testing is a standard approach; however, its inclusion in our study provides a critical foundation for comparing AZT-AVI activity with other β-lactam/β-lactamase inhibitor combinations and last-resort agents such as colistin and cefiderocol. The detailed MIC data, interpreted using EUCAST breakpoints, adds significant clinical relevance by contextualizing the efficacy of AZT-AVI against MDR pathogens, particularly in challenging infection scenarios.

Comment 5: I would like to see new and additional information that would bring it to a new level.

Reply: In alignment with this valuable suggestion, we have added a comprehensive phylogenetic analysis of the Klebsiella pneumoniae isolates, which highlights the prevalence of high-risk clones and their associated genomic traits. Additionally, we have expanded the resistome analysis to include detailed insights into antimicrobial resistance genes, plasmid content, and insertion sequences. These enhancements provide a deeper and more nuanced understanding of the genomic basis of resistance in MBL-producing Enterobacterales, elevating the scientific value of the study.

Round 2

Reviewer 5 Report

Comments and Suggestions for Authors

A-  It is good to see the changes to the manuscript.

B-  The authors made a good effort to enhance the importance of the topic.

C- I believe that with these changes and additions , the authors have elevated the  manuscript to a level that can be accepted.